# Robot-Assisted and Manual Cochlear Implantation: An Intra-Individual Study of Speech Recognition

**DOI:** 10.3390/jcm12206580

**Published:** 2023-10-17

**Authors:** Clémentine Maheo, Antoine Marie, Renato Torres, Jerrid Archutick, Jean-Christophe Leclère, Remi Marianowski

**Affiliations:** 1Department of Otolaryngology Head Neck Surgery, University Hospital Center, Hospital Morvan, 2 Avenue Foch, 29200 Brest, France; antoine.marie@chu-brest.fr (A.M.); jean-christophe.leclere@chu-brest.fr (J.-C.L.); remi.marianowski@chu-brest.fr (R.M.); 2Technologies et Theérapie Génique Pour la Surdité, Institut de l’Audition, Institut Pasteur/Université de Paris Cité/INSERM, 63 rue de Charenton, 75012 Paris, France; victor.torres-lazo@pasteur.fr; 3Medicine Department, School of Medicine, University of Limerick, V94 T9PX Limerick, Ireland; archutic@ualberta.ca

**Keywords:** electrode array insertion, robot-assisted implantation, hearing outcome, scala translocation

## Abstract

Cochlear implantation (CI) allows rehabilitation for patients with severe to profound hearing impairment. Although the use of a robotic assistant provides technical assistance to the surgeon, the assessment of the impact of its use on auditory outcomes remains uncertain. We aim to compare the hearing results of patients who underwent bilateral cochlear implantation; one side was performed with manual insertion and the other side with robot-assisted insertion. The electrode array intrascalar positioning and the surgery duration were also studied. This retrospective intra-individual study involved 10 patients who underwent bilateral cochlear implantation. The study included two infants and eight adults. The unique composition of this cohort enabled us to utilize each patient as their own control. Regarding speech disyllabic recognition, pure tone average, ECAP, ratio of array translocation, basilar membrane rupture, and percentage of translocated electrodes, there was no difference between manual and robot-assisted CI groups. This study is the first to compare intra-individual hearing performance after cochlear implantation, either manually or robot-assisted. The number of patients and the time delay between manual and robotic implantation may have led to a lack of power, but there was no apparent difference in hearing performance between manual and robotic implantation.

## 1. Introduction

Cochlear implantation allows rehabilitation for patients with severe to profound hearing impairments. It can be used for both adults and children with syndromic and non-syndromic deafness [1]. It may improve speech perception by 80% as well as hearing level in pure tone audiometry [2]. For normal-hearing people, binaural hearing offers sound localization, speech intelligibility, and a better understanding of speech in noise. Since 2018, bilateral cochlear implantation has become a standard of care with reimbursement in Europe, the United States, and Canada. It improves speech perception in quiet and noise, sound localization, and music appreciation. For children, binaural hearing has a positive effect on speech and language development [3].

The proximity of the electrodes to the ganglion cells, the electrode array insertion depth, and the absence of electrode array translocation [4,5] are critical parameters for better hearing outcomes. The preservation of residual hearing is enhanced by the protection of the basilar membrane provided by a correct angle of insertion. Manual cochlear implantation can be affected by involuntary movements such as tremors, overshooting or undershooting, which can affect the precision of the insertion even for a senior otologist [6]. Several robots have been developed and used for cochlear implantation, such as the HEARO^®^ (CAScination AG, Bern, Switzerland), which provides autonomous, minimally invasive access to the inner ear by drilling a path through the temporal bone [7]. During the same period, a robotic assistant for cochlear implantation was designed, i.e., the RobOtol^®^ (Collin ORL, Bagneux, France), to reduce surgical trauma during the insertion of electrode arrays. This robotic assistant allows slower progression and less traumatic electrode positioning [8].

Elaborated since 2006 by UMR-S 1159 [9] and Collin© Company, the RobOtol^®^ was commercialized in 2016 and has been used in Europe and China since 2019. It is indicated for cochlear implantation, but can also be used for otosclerosis surgery or as an endoscope holder for various middle ear procedures [10]. Positioning the electrode array requires highly precise dexterity and, therefore, a significant learning curve. The benefits of this tool are improved precision of movement and the elimination of essential tremors [11]. Even with a narrow mastoid in the pediatric population, it can be used in children over one year of age [12]. Although the use of this robot provides technical assistance to the surgeon, the assessment of the impact of its use on auditory outcomes remains uncertain due to the inter-individual variability of preoperative audiometric thresholds and the various possible intrascalar positions of the electrode array.

We aim to compare the hearing results of patients who underwent bilateral cochlear implantation; one side was performed with manual insertion and the other side with robot-assisted insertion. The electrode array intrascalar positioning and the surgery duration were also studied. 

## 2. Materials and Methods

### 2.1. Patients and Study Design

This retrospective intra-individual study involved 10 patients who underwent bilateral cochlear implantation at a tertiary referral center. The study included 2 infants and 8 adults. Manual cochlear implantations were performed between January 2006 and June 2020, while robot-assisted implantations took place between September 2020 and May 2022. Sequential implantation was applied to 9 of the patients, while 1 patient underwent both manual and robot-assisted cochlear implantations in a single procedure. The unique composition of this cohort enabled us to utilize each patient as their own control. The local ethics committee approved the study (Brest CHRU: 29BRC23.0107; 22 June 2023). All patients consented to the use of their data. The study followed the French General Data Protection Regulation (GDPR). 

### 2.2. Cochlear Implants

Each patient received the same model on both ears. Three types of electrode arrays were used: Advanced BionicsTM HiFocus Slim J (*n* = 4). This is a straight electrode array with an active part of 20 mm, including 16 electrodes.Oticon/Neurelec^TM^ NeuroZTi (*n* = 4). This is a straight electrode array with an active part of 25 mm, including 20 electrodes.Med-EL Flex (*n* = 12). This is a flex-tip straight electrode array with an active part of 20, 24, or 28 mm, including 12 electrodes.

### 2.3. Radiological Analysis

The position of the electrode array was studied. Pre-implantation computed tomodensitometry (CT) and post-implantation CT (within 24 h after surgery) were performed in all patients. Using 3D multiplanar reconstruction of the images with Horos v.2.2.0 open-source software (https://horosproject.org/, accessed on 14 September 2023), we determined the intrascalar position of each electrode according to the basilar membrane [13]. They have been classified as “all in ST” when they were located in the scala tympani, “translocated” if they were partly or completely in the scala vestibuli, and “BMR” in cases of basilar lamina rupture. 

### 2.4. Electrically Evoked Compound Action Potential (ECAP)

The electrically evoked compound action potential (ECAP) was measured intraoperatively. It is a marker of auditory nerve function after cochlear implant surgery. The ECAP represents a synchronous response from electrically stimulated auditory nerve fibers. It is used intraoperatively to confirm auditory nerve, device integrity, and electrode functionality and postoperatively by audiologists for speech processor programming [14]. 

### 2.5. Audiological Evaluations

We performed behavioral audiometry for the infants and aided hearing evaluation for the adults before implantation, at 3 months and 1 year postoperatively for each side. Pure-tone audiometry was assessed to determine the hearing thresholds at the frequencies 0.25, 0.5, 0.75, 1, 2, 4, and 8 kHz. The pure-tone average (PTA) was calculated as the mean of the thresholds at 0.5, 1, 2, and 4 kHz. A low-frequency PTA (0.25–0.5–0.75 kHz) and a high-frequency PTA (4–8 kHz) were calculated.

Speech perception in silence using disyllabic word lists was performed before surgery, at 3 months, and 1 year postoperatively. The speech discrimination score (SDS) was determined at 60 dB HL and expressed as a percentage. 

### 2.6. Surgical Technique 

All interventions were performed by a senior otologist using the same surgical approach for both insertion techniques. A retro auricular approach, a regular transmastoid route with posterior tympanotomy, and exposure of the round window were performed. Manual or robot-assisted insertion was applied by one right-handed surgeon via a round window approach. The cochlear implant was placed in a sub-periosteal area, with or without the drilling of a dedicated bed in the temporal bone. Robot-assisted electrode array (EA) insertion was performed using the RobOtol^®^ system as a platform. The surgeon controlled the robotic arm using the Space Mouse with the right hand (3D-connection, Waltham, MA, USA) mounted on the rail of the operating table. After adjusting the robotic arm to the optimum position and aiming it at the ideal insertion axis with the high speed, 10 mm/s max, the EA was introduced slowly into the cochlear through the round window with the low-speed mode, 1 mm/s max, advancing to the target position without interruption and then released carefully. The same standard technique was used to close the surgical cavity on both sides with autogenous cranial bone pate obliteration, after which the wound was closed layer by layer.

### 2.7. Installation and Surgery Durations

We conducted a comparison between the manual and robot-assisted methods to assess the surgical time. The duration of the following procedures was meticulously recorded: The additional preparation time of the RobOtol**^®^**: space mouse fixation, robotic arm positioning, and sterile draping of both.The EA preparation for robotic assistance and insertion time: mounting the EA on the robotic tool, aiming the robotic arm in the right axis, and EA robotic insertion time.

### 2.8. Statistical Analysis 

Statistical analyses were performed using SPSS 25 software. Non-parametric tests were performed to assess the association between hearing performance and robotic/manual insertion. Categorical and quantitative paired variables were analyzed using the Fisher’s exact test and the Wilcoxon signed rank test, respectively. Quantitative unpaired variables were analyzed using the Mann–Whitney test. No regression models were used for missing data. Data were saved onto our hospital’s database server using a password-protected spreadsheet per the Data Protection Act. The significance threshold was set at 0.001, and the suggestiveness threshold was set at 0.05 [15,16].

## 3. Results

No difference was noted between both groups: side (left/right), preoperative pure tone average, and speech discrimination at 60 dB HL of the implanted ear with a hearing aid (Table 1). There was a suggestive difference between groups in age. Both sides were implanted with the same electrode array type (Table 2), and the mean delay between both sides was 39(3,25) ± 51(4,25) months [0–178(14,83)]. There were no post-operative complications such as facial palsy or local infection. 

Speech disyllabic recognition shows comparable improvement between the manual and robot-assisted groups at the 3-month post-operative assessment (26.3% vs. 31.3%, respectively; *p* = 0.74, Wilcoxon signed rank test; Table 3) and at the one-year post-operative assessment (67.5% vs. 66.23%, respectively; *p* = 0.93; Wilcoxon signed rank test; Table 3). Regarding the pure tone average at one year after surgery, there was no significant difference between groups for low (*p* = 1), medium (*p* = 0.6), or high (*p* = 0.26) frequencies. 

Regarding ECAP, there was no difference between both groups (manual: 76.8%, robot-assisted: 83.1%; *p* = 0.67; Wilcoxon signed rank test; Table 3). 

The ratio of array translocation and basilar membrane rupture was not different between robot-assisted (*n* = 4, 40%) and manual (*n* = 2, 20%) groups (*p* = 1; Fisher’s exact test; Table 2). Regarding the percentage of translocated electrodes, there were no differences between manual (*n* = 15, 13%) and robot-assisted (*n* = 21, 15%) groups (*p* = 0,49; Wilcoxon signed rank test; Table 3). The ratio of array translocation did not differ according to the type of cochlear implant used (AB = 2/6, Med-El = 4/7, Oticon = 2/8; *p* = 1; Fisher’s exact test; Table 3). 

Array translocation shows no association with impaired speech perception in silence at one year (translocation: 64 ± 37.8% *n* = 6, no translocation: 64.4 ± 27.43% *n* = 12; *p* = 1; Mann–Whitney test). Moreover, no correlation was observed between array translocation and an impaired postoperative low PTA (*p* = 0.36), medium PTA (*p* = 0.81), or high PTA (*p* = 0.27) at one year (*p* = 0.42; Mann–Whitney test; Table 3). There was no association between the absence of ECAP in at least 1 electrode pair and worse speech recognition at 1 year (*p* = 0.24; Mann–Whitney test; Table 3). Moreover, array translocation was not associated with the absence of ECAP in at least 1 electrode pair (*p* = 0.50; Mann–Whitney test; Table 3). 

When using RobOtol^®^, the mean additional preparation time was 4 ± 1 min [3,4,5] and the mean insertion time was 11 ± 5 min [6,7,8,9,10,11,12,13,14,15,16,17,18,19,20,21]. Concerning surgery duration, there was no difference between manual (95 min) and robot-assisted (109 min) groups (*p* = 0.24; Wilcoxon signed rank test; Table 2). All electrode arrays used were straight; there was no difference between the three types regarding the insertion time (*p* = 0.26) and surgery duration (*p* = 0.19; Mann–Whitney test; Table 2). 

## 4. Discussion

In this intra-individual study, hearing outcomes after robot-assisted or manual cochlear implantations were similar for speech in silence at 3 months and 1 year after surgery. 

There were similar pure-tone audiometry thresholds with robot and manual electrode array insertions for all frequencies. Another study reported the similar hearing performance of 42 deaf adults implanted either manually or robot-assisted [17]. In this study conducted by Torres et al., the participants were paired by age and electrode array type. The same robot was used in both studies. The mean age of the cohort in Torres et al. was 55 years, whereas the population in our study was 42 years.

The RobOtol^®^ preparation time reported in this study (210 ± 56 s) was comparable to the additional preparation time in another study performed in five children and one adult with simultaneous bilateral CI implantation with robotic assistance on one side and manual insertion on the other side (208 ± 106 s) [12]. They also reported the time for positioning the EA on the robotic tool, opening the round window, aiming the robotic arm along the insertion axis (242 ± 124 s), and the EA insertion time with robotic assistance (198 ± 65 s). The duration of insertion under robotic assistance was lower than that by manual insertion (73 ± 10 s) in their study. Our results for combined EA preparation and insertion duration combined by RobOtol^®^ (630 ± 301 s) appeared to be slightly longer than their study. This can be explained by the learning curve effect, as most patients in our study underwent surgery early after the RobOtol^®^ acquisition. The insertion time tended to decrease with each procedure, except for one patient who had a particularly complex insertion. Unfortunately, the manual insertion time was not reported in our retrospective study. However, in our study, the duration of surgery was not significantly different between the manual and robotic insertion groups. This data were not previously reported in other studies.

In our study, there was no difference in insertion time or surgery time with the type of electrode array, including the Med-El FLEX-tip electrode array. It was also important to note that the Neuro ZTI electrode array was less rigid than the SlimJ electrode array. In the study conducted by Jia et al., they also reported a longer preparation time for the electrode array for robotic assistance when using the FLEX^SOFT^ electrode array. In terms of insertion times, they reported a shorter insertion time with the pre-curved perimodiolar electrode array than with FLEX^SOFT^ and another straight electrode array [12]. 

The literature is rare regarding the relationship between translocation and poorer speech recognition. Earlier studies reported better speech performance when the electrode array was fully inserted into the scala tympani [18,19]. More recently, the results of another study were more consistent with our findings [17]. Their study also reported an association between a proximal translocation and a decrease in the high-frequency pure tone average. This association was not found in our study. The presence of a translocation or basilar membrane rupture, regardless of its location, did not affect the pure tone average at one year post-operatively. 

The pre-curved electrode array had a pre-determined shape that may not match the variable coiling pattern of individual cochleae [20,21], and there was a wire to bring the pre-curved electrode into a straight configuration [22]. Therefore, the pre-curved electrode was more likely to cause trauma and translocation than a straight electrode. 

Ketterer et al. reported the highest rate of primary scala vestibular insertions with the Contour Advance^®^ pre-curved electrode and the highest rate of secondary translocations from the scala tympani with the FLEX^SOFT^ electrode. With the FLEX^24^ and FLEX^28^ also used in our study, they reported less than ten percent of secondary translocation for the FLEX^24^ inserted in the RW and approximately ten percent of primary scala vestibular insertion or secondary translocation with the FLEX^28^ [23]. In our small cohort of the FLEX^20,24,28^ electrode arrays, we counted sixteen percent of translocations, excluding isolated basilar membrane ruptures.

Our results are consistent with several studies on ECAP. A systematic review has summarized the literature on the use of intraoperative ECAP as a predictor of speech perception and found no clear relationship between ECAP and cochlear implant performance [24]. Cosetti et al., reported no statistically significant differences in cochlear implant performance between patients with absent ECAP in one or more electrodes compared to those with measurable ECAP in all electrodes described [25]. Cooper et al. characterized a series of cochlear implant patients with completely absent intraoperative ECAP and concluded that it did not predict poor cochlear implant performance [26]. 

The mean delay to perform the second side implantation was 39 ± 51 months [0–178]. The study conducted by Baron et al. on pediatric populations compared two groups according to the inter-CI interval: < 36 months for the “early group” versus ≥ 36 months for the “late group” [27]. Using these criteria, more than half of the patients in our study were implanted late in the second ear.

The length of hearing loss before the second cochlear implant was a factor negatively correlated to speech understanding [28]. Regarding the interval between implantations, the literature is inconsistent. Some studies showed a longer interval did not necessarily lead to poor results for the second implanted side [27,28]. Sequential bilateral CI significantly improved speech intelligibility and perception in patients with both early and late second CI [27]. Another study reported no effect on the interval between the implants on sound localization [29]. However, other studies conducted by Zeitler et al. and Steffens et al. reported negative correlations between the inter-CI interval and speech perception in noise [30,31]. 

To note, Zeitler et al. reported that speech perception continued to improve, whatever the inter-CI interval [31]. 

This intra-individual study overcomes the variability caused by the etiology of the hearing loss and the type of implant used. However, it has some limitations. This study cannot exclude variability due to the duration of profound deafness; only one patient underwent synchronous implantation, which may have biased the results in favor of robotic implantation in the others. A large multi-center, prospective, single-blind, randomized trial will be needed to ensure that there is no effect of robot-assisted insertion on hearing outcomes. A prospective intra-individual study with simultaneous implantation (one side with robotic assistance and manual insertion on the other) would be another option.

## 5. Conclusions

This study is the first to compare intra-individual hearing performance after cochlear implantation, either manually or robot-assisted. The number of patients and the time delay between manual and robotic implantation may have led to a lack of power, but there was no apparent difference in hearing performance between manual and robotic implantation. 

There was no significant difference between the two methods in terms of operative time or the incidence of array translocation or basilar membrane rupture. No complications were observed. This confirms the overall safety and reliability of both manual and robotic approaches to cochlear implantation.

## Figures and Tables

**Table 1 jcm-12-06580-t001:** Clinical characteristics.

	Robot-Assisted	Manual	*p*-Value
Age (years ± DS [min–max])	44 ± 27 [1–73]	40 ± 26 [1–70]	0.014
Side (Left/Right)	5/5	5/5	1
Preoperative PTA ^1^ (dB SL)—implanted ear	100.8 ± 25.9	109.48 ± 9.37	0.90
Preoperative SDS 60 dB ^2^ (%)—implanted ear	12.5 ± 23.75	0 ± 0	0.37

^1^ PTA: pure tone average; ^2^ SDS 60 dB: speech discrimination percentage at 60 dB. Data are expressed as mean ± SD [min–max].

**Table 2 jcm-12-06580-t002:** Cochlear implant types and surgical data for the ten cases presented in this study.

**Sex**	**Age at Surgery (Years)**	**Model**	**Type**	**Side**	**Surgical Technique**			
					**Surgery ** **Duration (min)**		**RobOtol^®^ ** **Preparation Time (min)**	**RobOtol^®^EA ** **Preparation/** **Insertion Time (min)**
					**Manual**	**Robot-Assisted**		
M	57	AB ^1^ SlimJ	Manual	Right	88			
	58	AB SlimJ	Robot	Left		148	N/A	N/A
F	70	Oticon	Manual	Left	90			
	73	Oticon	Robot	Right		100	3	21
F	28	Med-El	Manual	Left	N/A			
	33	Med-El	Robot	Right		70	3	8
F	50	Oticon	Manual	Right	90			
	52	Oticon	Robot	Left		80	3	10
F	1	Med-El	Manual	Right	77			
	1	Med-El	Robot	Left		N/A	N/A	N/A
M	1	Med-El	Manual	Left	120			
	2	Med-El	Robot	Right		115	3	8
F	61	Med-El	Manual	Right	N/A			
	65	Med-El	Robot	Left		120	3	7
F	64	Med-El	Manual	Left	120			
	67	Med-El	Robot	Right		146	2	11
F	30	Med-El	Manual	Left	N/A			
	45	Med-El	Robot	Right		137	4	15
M	66	AB SlimJ	Manual	Right	81			
	66	AB SlimJ	Robot	Left		62	5	6

^1^ AB: Advanced Bionics^®^.

**Table 3 jcm-12-06580-t003:** Scalar position and speech disyllabic recognition.

Sex	Age at Surgery	Implantation Type	EA Positioning		Speech Perception at 60 dB (%)		PTA ^1^ at 1 Year (dB)		
			Translocation (% EA)	ECAP ^2^	3 Months	1 Year	Low-Frequency	Mid-Frequency	High-Frequency
M	57	Manual	All in ST	16/16	50	70	37.5	35	50
	58	Robot	Distal translocation (25%)	16/16	30	100	32.5	30	40
F	70	Manual	All in ST	4/20	80	90	35	37.5	45
	73	Robot	Distal (20%)	4/20	90	100	40	40	40
F	28	Manual	Proximal BMR ^3^ (30%)	10/12	10	60	35	35	42.5
	33	Robot	Translocation (100%)	12/12	N/A	10	32.5	35	35
F	50	Manual	All in ST	20/20	30	90	37.5	37.5	45
	52	Robot	All in ST	19/20	80	100	40	35	45
F	1	Manual	Artifacts	11/12	N/A	N/A	55	50	55
	1	Robot	All in ST	12/12	N/A	N/A	55	55	60
M	1	Manual	All in ST	8/12	N/A	N/A	37.5	37.5	45
	2	Robot	Distal translocation (8%)	12/12	N/A	N/A	35	42.5	45
F	61	Manual	Proximal BMR (70%)	12/12	20	50	37.5	32.5	35
	65	Robot	All in ST	12/12	20	50	30	27.5	32.5
F	64	Manual	All in ST	12/12	10	30	27.5	32.5	35
	67	Robot	All in ST	7/12	0	20	37.5	40	40
F	30	Manual	Artifacts	10/12	N/A	80	N/A	N/A	N/A
	45	Robot	All in ST	7/12	20	N/A	N/A	N/A	N/A
M	66	Manual	All in ST	4/16	10	70	35	32.5	40
	66	Robot	All in ST	16/16	10	60	37.5	32.5	35

^1^ PTA: pure tone average (dB SL); ^2^ ECAP: electrically evoked compound action potentials; ^3^ BMR: basilar membrane rupture. The array translocation was expressed as a percentage.

## Data Availability

The data presented in this study are available on request from the corresponding author. The data are not publicly available due to privacy restrictions.

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
