# Peer review of "Robot-Assisted and Manual Cochlear Implantation: An Intra-Individual Study of Speech Recognition"

_jcm, 2023, doi:10.3390/jcm12206580_

Round 1
Reviewer 1 Report
This is an very interesting study on the interest of robotic assisted insertion of a cochlear implant array although it is a retrospective study with a limited number of cases.
Nevertheless, it’s an intra-individual study which allow to prevent the variability caused by the etiology of the hearing loss and the type of implant used. So, no difference was noted between both groups: side(left/right), preoperative pure tone average, and speech discrimination at 60 dB HL of the implanted ear with a hearing aid.
The authors conclude that there was no apparent difference in hearing performance between manual and robotic implantation, as well no difference of array translocation and basilar membrane rupture.
- I have one minor comment on the statement l62 : « less invasive access to the cochlea ; and a reduction of the learning curve »
I don’t agree with this statement because you need with the robot an even more mastoidectomy to perform the insertion ; and the learning curve of robotic use is as long as a manual insertion. So please moderate this affirmation.
- Note also a typo error l 202 (line wrapped) : In this study conducted by
Author Response
Dear reviewer, thank you very much for taking the time to review this manuscript. Please find the detailed responses below
- I have one minor comment on the statement l62 : « less invasive access to the cochlea ; and a reduction of the learning curve »
I don’t agree with this statement because you need with the robot an even more mastoidectomy to perform the insertion ; and the learning curve of robotic use is as long as a manual insertion. So please moderate this affirmation.
Thank you for pointing this out. We agree with this comment, the wording of these elements is indeed incorrect and we have removed it. ‘The benefits of this tool are improved precision of movement and elimination of essen-tial tremors.’
- Note also a typo error l 202 (line wrapped) : In this study conducted by
Thank you, typo error has been corrected
Reviewer 2 Report
Thank you for the opportunity to review this interesting manuscript with a comparison between the robot-assisted and manual cochlear implantation.
Abstract: Clear and well written.
Introduction: Good introduction to the subject investigated.
Material and Methods: Well written and structured, and detailed description of the materials and methods for this study. Interesting approach of this cohort where each patient is their own control.
Results: Interesting study showing no differences between manual and robot-assited method in CI groups studied. Tables are clear.
Discussion: The discussion is well discussed and written.
Conclusions: Conclusions could be written with some more details about findings. Even non-significant findings are important.
References: Good and relevant references both from earlier and recent publications.
Author Response
Dear reviewer, thank you very much for taking the time to review this manuscript. Please find the detailed responses below
Abstract: Clear and well written.
Introduction: Good introduction to the subject investigated.
Material and Methods: Well written and structured, and detailed description of the materials and methods for this study. Interesting approach of this cohort where each patient is their own control.
Results: Interesting study showing no differences between manual and robot-assited method in CI groups studied. Tables are clear.
Discussion: The discussion is well discussed and written.
Conclusions: Conclusions could be written with some more details about findings. Even non-significant findings are important.
References: Good and relevant references both from earlier and recent publications.
For the conclusion, we have added the following paragraph: “There was no significant difference between the two methods in terms of opera-tive time and the incidence of array translocation and basilar membrane rupture. No complications were observed. This confirms the overall safety and reliability of both manual and robotic approaches to cochlear implantation.”